# Accuracy of the Sentence-BERT Semantic Search System for a Japanese Database of Closed Medical Malpractice Claims

**Naofumi Fujishiro \*** **, Yasuhiro Otaki and Shoji Kawachi**

General Medical Education and Research Center, Teikyo University, Tokyo 173-8605, Japan
\* Correspondence: naofuji@med.teikyo-u.ac.jp

**Abstract:** In this study, we developed a similar text retrieval system using Sentence-BERT (SBERT) for our database of closed medical malpractice claims and investigated its retrieval accuracy. We assigned each case in the database a short Japanese summary of the accident as well as two labels: the category was classified as a hospital department mainly, and the process indicated a failed medical procedure. We evaluated the accuracies of a similar text retrieval system with the two labels using three different multilabel evaluation metrics. For the encoders of SBERT, we employed two pretrained BERT models, UTH-BERT and NICT-BERT, that were trained on huge Japanese corpora, and we performed iterative optimization to train the SBERTs. The accuracies of the similar text retrieval systems using the trained SBERTs were more than 15 points higher than those of the Okapi BM25 system and the pretrained SBERT system.

**Keywords:** closed claims; medical malpractice; BERT; Sentence-BERT; BM25; information retrieval; natural language processing; multilabel machine learning



## 1. Introduction

To prevent the recurrence of medical accidents, it is important to learn from past accidents. In Japan, the Project to Collect Medical Near-Miss/Adverse Event Information operated by the Japan Council for Quality Health Care [1] and the Medical Accident Investigation System operated by the Japan Medical Safety Research Organization (Medsafe Japan) [2] collect and analyze data on medical accidents that meet certain requirements and conduct educational activities to prevent the recurrence of similar medical accidents. The Japan Council for Quality Health Care created a database of information on medical accidents for their near-miss and adverse event project and made a part of the information publicly available, and medical institutions use this database for risk management by searching for similar medical accidents using keywords. However, because medical terminology is not uniform, it is not yet possible to fully derive similar medical accident cases through keyword searches.

One source of medical accident case studies is insurance companies' closed claims. A representative example is the Data Sharing Project of the MPL Association, in which ~70% of MPL (medical professional liability) insurers in the United States participate. The closed claims collected through this project are analyzed from various aspects and used for medical institutions' risk management [3].

In Japan, Sompo Japan Insurance Inc. (SJ) holds ~70% of the MPL insurance market. We have analyzed the SJ closed claims and reported on their usefulness as a source of information on medical accidents and medical disputes [4–6]. Currently, we are creating a database of closed SJ claims including textual information for more efficient analysis. Furthermore, we are also developing not only a keyword search system but also a similar case search system with higher accuracy for this database. In the future, we hope to return these results to medical institutions so they can utilize them in their risk management.

To effectively utilize our closed-claims database, it is essential to have a system to extract past medical accidents that are similar to the one being investigated. Because our

database contains textual information, we considered that a similar document retrieval system would be useful. There are two main types of similar document retrieval systems: lexical similarity and semantic similarity. A lexical similarity system matches keywords or phrases in a search query to text content; it focuses on exact word matching without considering the context or the meanings of words. A representative system based on lexical similarity is Okapi BM25 [7], whose implementation began in the 1990s; it is based on term frequency–inverse document frequency (TF-IDF) and is still widely used today.

In contrast, a semantic similarity system focuses on the meaning and context of the words in a query, considering synonyms, related concepts, and the context of a word's usage. This type of search produces more relevant results by understanding the intent behind an input query. Retrieval systems based on semantic similarity are built using machine learning models that output word-embedding vectors. Representative models are Word2Vec [8], Doc2Vec [9], Glove [10], fastText [11], BERT [12], and GPT-3 [13] in the order of publication. Among these models, we chose to employ BERT (bidirectional encode representations from transformers) because of the large numbers of research publications in the fields of medicine and Japanese natural language processing, as well as available frameworks and libraries.

BERT is a pretrained language model that can be fine-tuned for a variety of natural language processing (NLP) tasks. The algorithm uses an encoder–decoder architecture with a transformer-based encoder. The first key feature of the BERT algorithm is bidirectional training. BERT models are trained on a large corpus using a bidirectional approach where the model can consider left-to-right as well as right-to-left context. This allows the model to understand the meaning of words based on the surrounding context. The second key feature is the Masked Language Model. During pretraining, a certain percentage of input tokens are randomly masked, and the model is trained to predict the original tokens based on the surrounding context. This forces the model to learn a deeper understanding of the context. Then, the pretrained BERT model can be fine-tuned on a specific NLP task, such as sentiment analysis, question answering, or named entity recognition, by adding an additional output layer and training the model on a smaller dataset. The feature of only requiring a small dataset is suitable for this study without a large dataset.

Among the various derivatives of BERT, Sentence-BERT (SBERT) [14] is highly regarded for its search accuracy and speed. Therefore, we decided to develop a similar accident retrieval system for our closed-claims database using SBERT and to investigate its retrieval accuracy.

## 2. Materials and Methods

### 2.1. BERT

In this study, we adopted SBERT as a semantic search system to find similar medical accident summaries for our Japanese database of closed medical accident claims. To implement SBERT, we used Sentence-Transformers [15], a library in the Python language. This library is based on PyTorch, a deep learning framework, and the Transformers repository of Hugging Face, Inc.

As shown in Figure 1, there are two types of SBERT: the bi-encoder model, which uses two BERT encoders to compare two texts, and the cross-encoder model, which uses only one BERT encoder. The bi-encoder model has the advantage of operating at high speed, but the cross-encoder model is more accurate and is suitable for re-ranking small numbers of documents [15]. Because our final goal was to eventually be able to find similar texts from more than 30,000 texts in the future, we chose the bi-encoder model, which is suitable for this application.

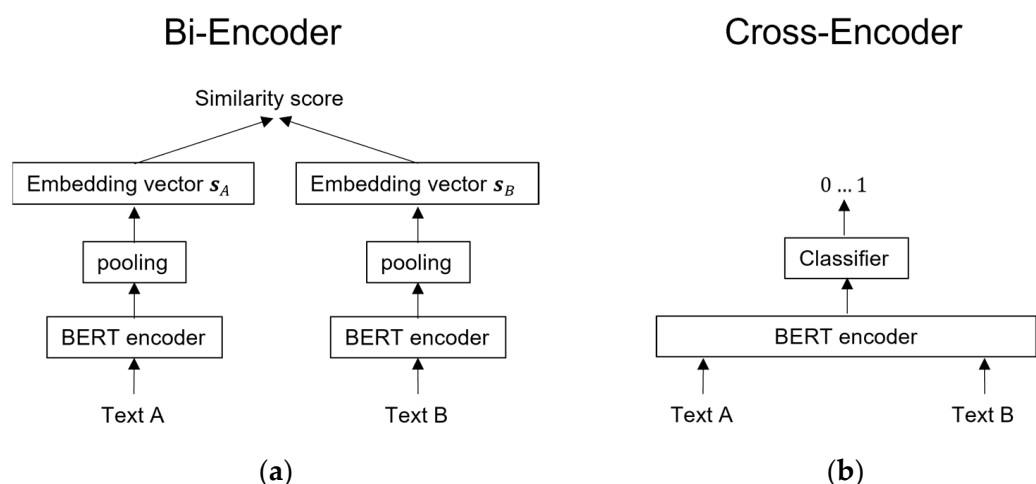

**Figure 1.** SBERT architectures: (**a**) the bi-encoder model and (**b**) the cross-encoder model.

We employed the two Japanese pretrained BERT models shown in Table 1 for the SBERT encoders. UTH-BERT [16] was trained with ~120 million lines of clinical texts stored in the electronic health record system of the University of Tokyo Hospital (UTH). Although UTH-BERT does not have a large vocabulary (25,000 tokens), it is rich in medical terms. Therefore, we considered UTH-BERT to be suitable for this study of medical accidents.

**Table 1.** The specifications of the pretrained BERT models.

|  |  | UTH-BERT | NICT-BERT |
|---|---|---|---|
|  | Publisher | The University of Tokyo Hospital | The National Institute of Information and Communications Technology |
|  | Pretraining corpus | Clinical text (120 million) | All Japanese Wikipedia |
| Tokenizer | Morphological analyzer | MeCab | |
|  | External dictionary | Mecab-ipadic-neologd, J-MeDic | Jumandic |
|  | Maximum sequence length | 512 | |
|  | Number of vocabularies | 25,000 | 100,000 [1] |

[1] We used the BPE version.

In contrast, NICT-BERT [17] is a generic BERT model created by the National Institute of Information and Communications Technology (NICT) and trained on all articles in Japanese Wikipedia. Because it has a rich vocabulary of 100,000 tokens, we chose it for comparison with UTH-BERT. For both the UTH-BERT and NICT-BERT tokenizers, we used the morphological analysis engine MeCab [18] implemented in Hugging Face Transformers with the external dictionaries specified for each BERT model.

*2.2. Datasets of Medical Accidents*

We used 1165 closed claims handled by SJ's Tokyo office from April 2018 to March 2019 as the dataset for medical accidents. For each case, we manually created a short text summarizing the medical process, patient outcome, and legal liability outcome based on paper documents stored in the SJ office. Figure 2 shows a histogram of the number of tokens in the short texts; there is little difference between the histograms for the UTH-BERT and NICT-BERT tokenizers. The mean and maximum numbers of tokens were 123 and 399 for the UTH-BERT tokenizer and 126 and 425 for the NICT-BERT tokenizer, respectively. The maximum numbers of tokens were smaller than the maximum sequence length of 512 in both UTH-BERT and NICT-BERT, and no sentence truncation occurred.

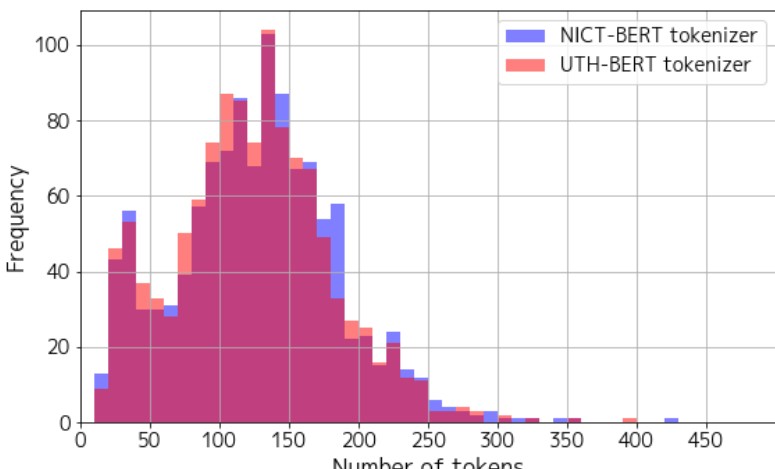

**Figure 2.** A histogram of the numbers of tokens in the short summary texts of medical accidents. The purple area is where the NICT-BERT tokenizer area shown in blue overlaps the UTH-BERT tokenizer area shown in red.

We also manually annotated each accident case with its category and process; the category identified the hospital department, medical profession, and property damage (e.g., patient's spectacles dropped and broken), and the process indicated a failed medical procedure. Figure 3 shows the different categories and processes and the frequencies of appearance of each in the dataset. The most frequent categories were internal medicine, surgery, obstetrics and gynecology, orthopedics, and dentistry, and the most frequent processes were diagnosis, surgery, treatment, and management. There was no intentional bias in the data collection.

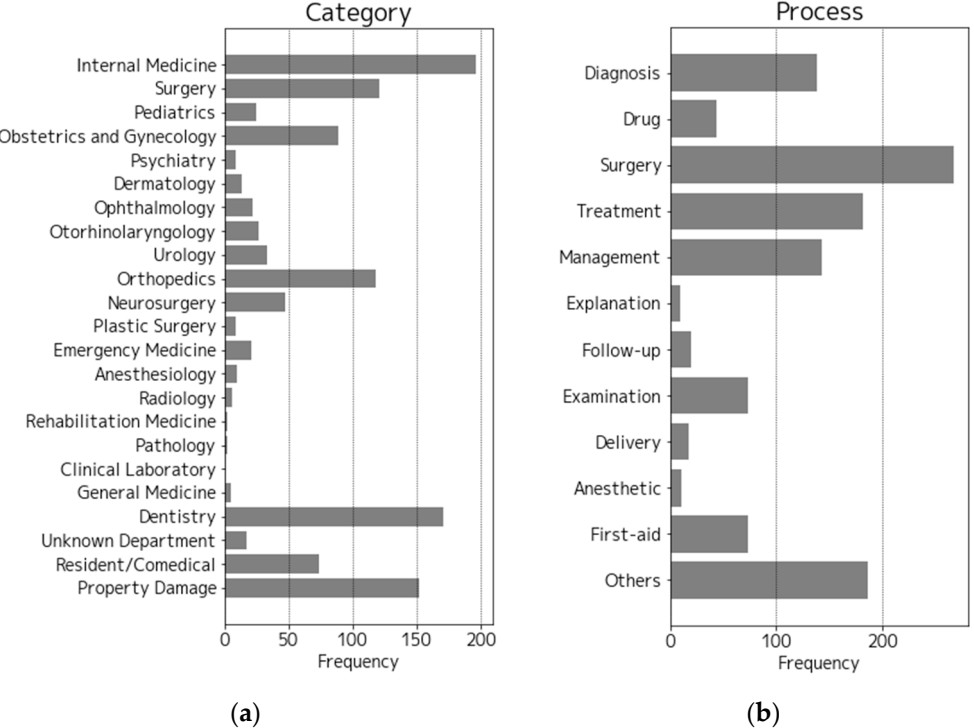

**Figure 3.** The (**a**) categories and (**b**) processes that occurred most often in the dataset according to their frequency of appearance.

Y. Otaki, a physician and lawyer, and an assistant on the project performed the documentation and annotation through trial and error from March 2019 to August 2022. Specific examples of the short texts and the labels are listed in in Tables 4 and 5 Section 3.2.

### 2.3. Similarity Scores

The similar text search engines we developed extracted the top-*K* texts with good similarity scores for a given query. Here let $\mathcal{D}$ be a multilabel dataset consisting of $|\mathcal{D}|$ multilabel instances $(s_i, Y_i), i = 1, 2, \cdots, |\mathcal{D}|$. The short text $s_i$ contains $M_i$ keywords $(w_{i,1}, w_{i,2}, \cdots, w_{i,M_i})$. In creating the keywords, we performed word segmentation using the UTH-BERT tokenizer or the NICT-BERT tokenizer and stop word removal using the dictionary of Japanese stop words from SlothLib [19]. The multilabel $Y_i$ is defined as a one-hot vector of $Y_i = (y_{i,1}, y_{i,2}, \cdots, y_{i,L})$, $y_{i,l} \in \{0, 1\}$, $1 \le l \le L$.

In similar text retrieval by SBERT, we measured the similarity between two texts with the Euclidean distance in the following formula:

$$euc\_dist(s_i, s_j) = \|\boldsymbol{s}_i - \boldsymbol{s}_j\| \tag{1}$$

where $\boldsymbol{s}_i$ is the embedding of the text $s_i$ derived by the BERT encoder. The smaller the Euclidean distance, the more similar the two texts.

We also built a similar text retrieval system using Okapi BM25 to compare with the SBERT system. For the Okapi BM25 system [7,20], we measured the similarity given by

$$BM25(\mathcal{D}, s_i, s_j) = \sum_{m=1}^{M_i} IDF(\mathcal{D}, w_{i,m}) \cdot \frac{f(w_{i,m}, s_j) \cdot (k_1 + 1)}{f(w_{i,m}, s_j) + k_1 \cdot \left(1 - b + b \cdot \frac{M_i}{\text{avgdl}}\right)}. \tag{2}$$

where $f(w_{i,m}, s_j)$ is the number of times that $w_{i,m}$ occurs in the text $s_j$, avgdl is the average text length in the dataset $\mathcal{D}$, and $k_1$ and $b$ are free parameters. $IDF(\mathcal{D}, w_{i,m})$ is the inverse document frequency weight of the query keyword $w_{i,m}$. This is given by

$$IDF(\mathcal{D}, w_{i,m}) = \log \frac{|\mathcal{D}| - n(\mathcal{D}, w_{i,m}) + 0.5}{n(\mathcal{D}, w_{i,m}) + 0.5}, \tag{3}$$

where $n(\mathcal{D}, w_{i,m})$ is the number of texts that contained $w_{i,m}$. A higher *BM25* indicates greater similarity between the two texts.

### 2.4. Evaluation Metrics

In this study, we employed the degrees of agreement between labels of a query and labels of similar texts as measures of similar text retrieval accuracy. We focused not on ranking the extracted similar texts but rather on how many of the top-*K* similar instances had labels that matched the query. In multilabel machine learning [21], there are three possible states of label agreement: complete match, partial match, and complete mismatch. First, we defined the exact match ratio as a measure of complete match. Here the top-*K* instances similar to a query instance $(s_q, Y_q)$ found by search engines in dataset $\mathcal{D}$ are $(s_k(\mathcal{D}, s_q), Z_k(\mathcal{D}, s_q)), k = 1, 2, \cdots, K$. The multilabel $Z_k(\mathcal{D}, s_q)$ is given by a one-hot vector of $Z_k(\mathcal{D}, s_q) = (z_{q,k,1}, z_{q,k,2}, \cdots, z_{q,k,L})$, $z_{q,k,l} \in \{0, 1\}$, $1 \le l \le L$. Then, the exact match ratio is given by

$$ExactMatchRatio(\mathcal{D}, (s_q, Y_q)) = \frac{1}{K} \sum_{k=1}^{K} 1(Y_q = Z_k(\mathcal{D}, s_q)), \tag{4}$$

where $1(x)$ is the indicator function: $1(x) = \{1 \text{ if } x \text{ is true, } 0 \text{ otherwise}\}$.

In some of the extracted similar texts, one label matches that of the query, but the other label is different (e.g., surgical vs. dental anesthesia accidents). Therefore, we defined the Hamming score in the following equation as a measure of partial match:

$$HammingScore\left(\mathcal{D}, \left(s_q, Y_q\right)\right) = \frac{1}{K} \sum_{k=1}^{K} \frac{\left|Y_q \cap Z_k\left(\mathcal{D}, s_q\right)\right|}{\left|Y_q \cup Z_k\left(\mathcal{D}, s_q\right)\right|}. \tag{5}$$

The Hamming score is sometimes referred to simply as accuracy in multilabel classification tasks; we also defined the Hamming loss as a measure of mismatch:

$$HammingLoss\left(\mathcal{D}, \left(s_q, Y_q\right)\right) = \frac{1}{KL} \sum_{k=1}^{K} \sum_{l=1}^{L} 1\left(y_{q,l} \neq z_{q,k,l}\right). \tag{6}$$

Finally, we averaged the above evaluation measures defined for individual queries for all queries. Let $\mathcal{D}^{train}$ be a training dataset and $\mathcal{D}^{test}$ be a testing dataset, and we defined the averaged metrics with the following equations:

$$AveragedExactMatchRatio\left(\mathcal{D}^{train}, \mathcal{D}^{test}\right) = \frac{1}{\left|\mathcal{D}^{test}\right|} \sum_{\left(s_q, Y_q\right) \in \mathcal{D}^{test}} ExactMatchRatio\left(\mathcal{D}^{train}, \left(s_q, Y_q\right)\right), \tag{7}$$

$$AveragedHammingScore\left(\mathcal{D}^{train}, \mathcal{D}^{test}\right) = \frac{1}{\left|\mathcal{D}^{test}\right|} \sum_{\left(s_q, Y_q\right) \in \mathcal{D}^{test}} HammingScore\left(\mathcal{D}^{train}, \left(s_q, Y_q\right)\right), \tag{8}$$

$$AveragedHammingLoss\left(\mathcal{D}^{train}, \mathcal{D}^{test}\right) = \frac{1}{\left|\mathcal{D}^{test}\right|} \sum_{\left(s_q, Y_q\right) \in \mathcal{D}^{test}} HammingLoss\left(\mathcal{D}^{train}, \left(s_q, Y_q\right)\right). \tag{9}$$

Larger averaged exact match ratios and Hamming scores and smaller averaged Hamming losses are better. In the results discussed below, *K*, the number of extracted similar instances, was fixed at 10.

### 2.5. Optimization Method

Table 2 lists the hyperparameters used when training SBERT. We used the loss function of multiple negatives ranking (MNR) loss [22] to train the SBERT used in the search engine. MNR loss has the advantage of only requiring a pair of original text and similar text, whereas the loss function of triplet loss used in the first SBERT paper [14] required a set of original text (anchor), similar text (positive), and dissimilar text (negative). For optimizer, batch size, learning rate, pooling strategy, and warmup rate, we followed the parameters used in the first SBERT paper. We adopted 15 as the optimal number of epochs based on preliminary experiments with other parameters fixed. The maximum sequence length was given by the maximum number of tokens in the dataset plus 3, the number of special tokens ([CLS] and [SEP]), and a trailing blank. We split the dataset into training, evaluation, and testing sets and allocated 80% to training, 10% to evaluation, and 10% to testing. We used random numbers for the data splitting and for the initial value setting during training. We calculated the evaluation metrics based on the means of the results with 10 different random seeds.

We note here that it is very labor intensive to manually create the text pairs needed to train SBERT with MNR loss; therefore, we developed the iterative method shown in Figure 4. First, as the zeroth iteration, a search engine using a pretrained SBERT and given the entire training dataset as a database corpus would find 10 similar texts for one query selected from the training dataset in order of best similarity score defined as Euclidean distance; here the same text as the query was excluded. If any of the 10 similar texts had labels that exactly matched those of the query, we created label-matched pairs with the query and the similar texts, and we repeated this with all texts in the training dataset as queries to build a dataset of label-matched pairs. Next, as the first iteration, we trained SBERT using the dataset of label-matched pairs with MNR loss. Then, we created a renewed dataset of label-matched pairs for the second iteration using a search engine with the trained

SBERT (first optimized SBERT in Figure 4). We performed these iterations 10 times. We expected that this iterative process would efficiently expand the dataset of label-matched pairs and gradually improve the accuracy of the search. Finally, we chose the SBERT model that yielded the best retrieval accuracy among the 10 iterations.

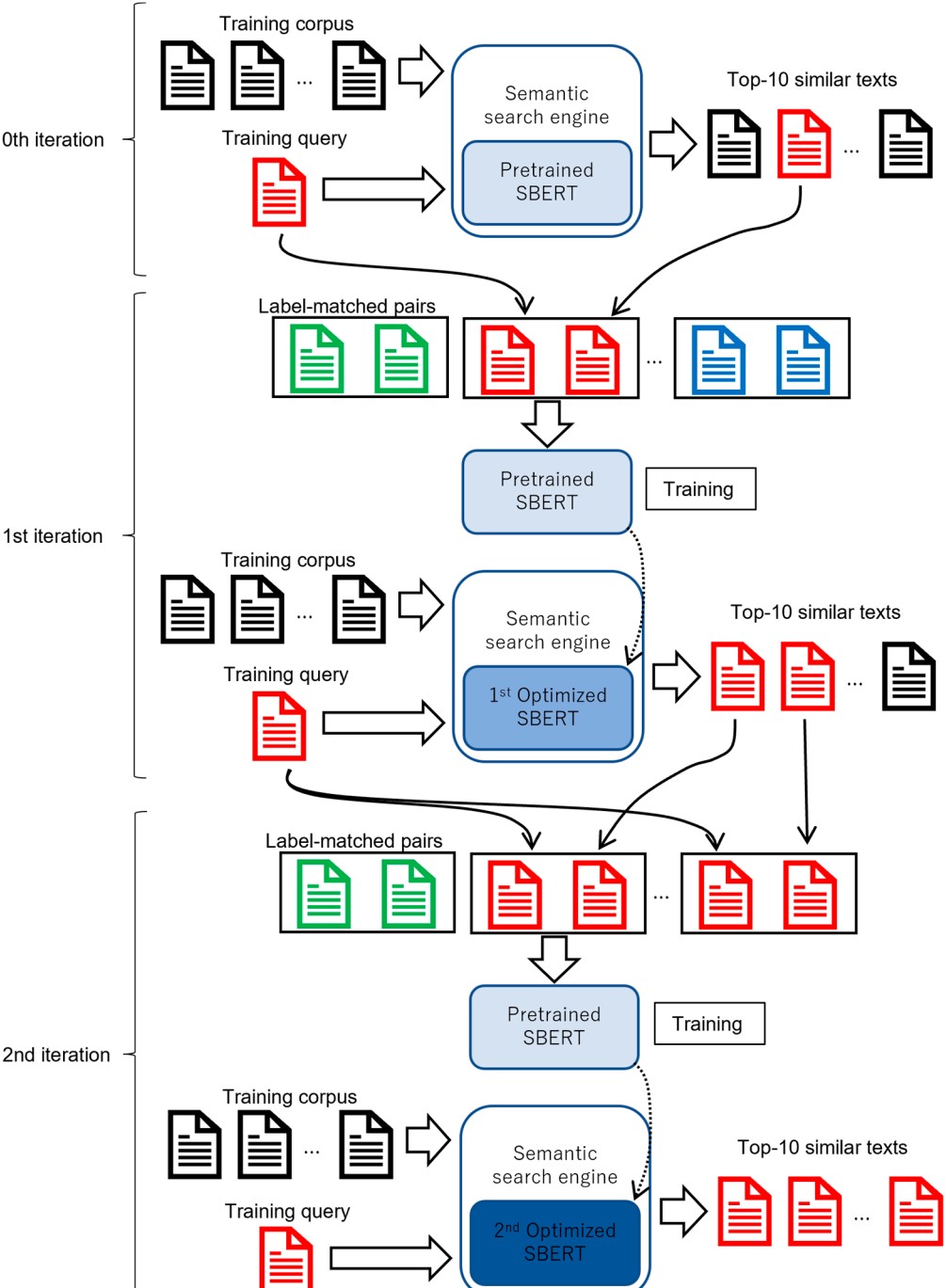

**Figure 4.** The SBERT iterative optimization method. Documents of the same color indicate that the labels are an exact match.

**Table 2.** Specific hyperparameters for SBERT training.

| Parameters | UTH-BERT | NICT-BERT |
|---|---|---|
| Loss function | Multiple negatives ranking loss | |
| Optimizer | Adam (warmup rate: 0.1) | |
| Pooling strategy | Mean | |
| Batch size | 16 | |
| Learning rate | $2 \times 10^{-5}$ | |
| Epochs | 15 | |
| Maximum sequence length | 399 + 3 = 402 | 425 + 3 = 428 |
| Split ratio of the dataset | Training:Evaluation:Testing = 0.8:0.1:0.1 | |
| Random seed | 1, 2, 5, 10, 20, 50, 100, 200, 500, 1000 | |

## 3. Results

### 3.1. Effectiveness of the Iterative Optimization Method

In this section, we present the results of training SBERT with the iterative optimization method described in Section 2.5. Figure 5a shows the number of label-matched pairs versus the number of iterations; as expected, the number of pairs showed a monotonic increase with the number of iterations and almost saturated at the eighth iteration. Figure 5b shows the averaged exact match ratio, a measure of complete match, versus the number of iterations; overall, UTH-BERT performed better than NICT-BERT. For UTH-BERT, the averaged exact match ratio was 27% with the pretrained model (zeroth iteration in the graph), but it improved to 45% at the fifth iteration, after which it declined. Figure 5c shows the averaged Hamming score, a measure of partial match, against the number of iterations, which showed the same characteristics as the change in the averaged exact match ratio. Figure 5d shows the averaged Hamming loss, a measure of mismatch, versus the number of iterations; a smaller averaged Hamming loss means fewer mismatches. Both UTH-BERT and NICT-BERT improved until the fourth iteration, after which we observed almost no change.

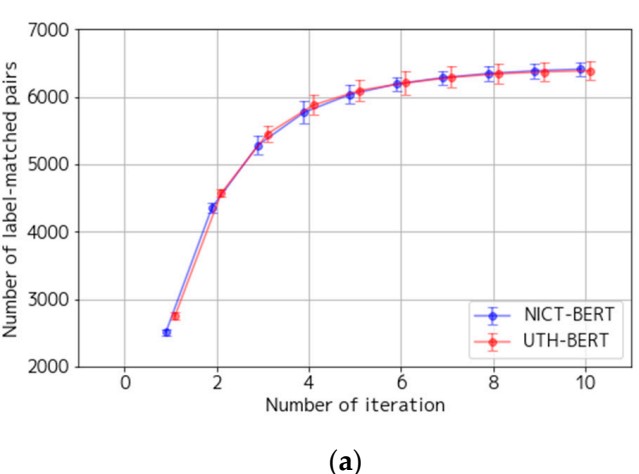

(**a**)

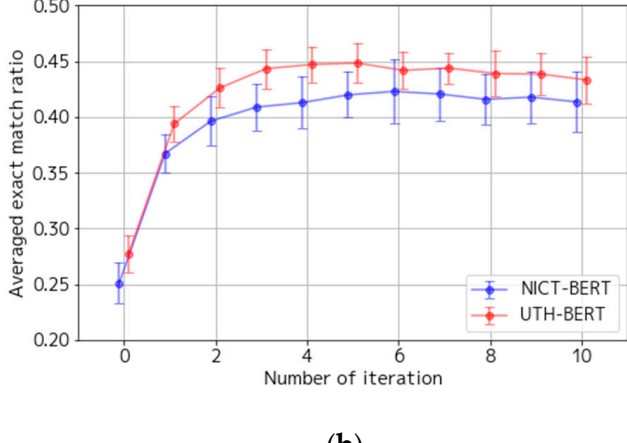

(**b**)

**Figure 5.** *Cont.*

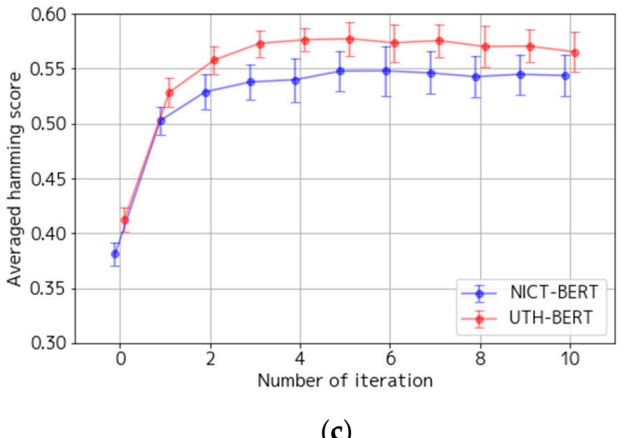

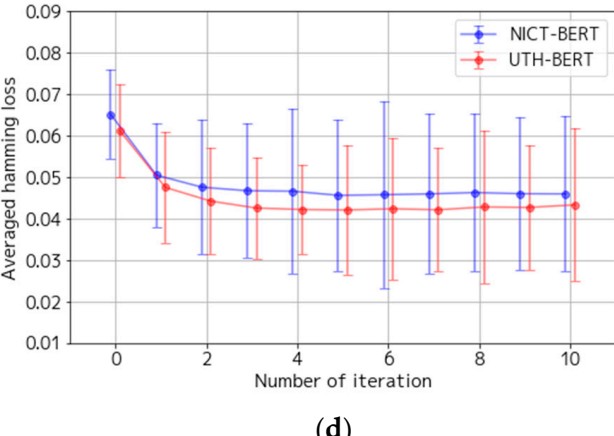

(**c**)                                                                                     (**d**)

**Figure 5.** Changes in the numbers of label-matched pairs and evaluation metrics with the optimization iterations: (**a**) number of label-matched pairs, (**b**) averaged exact match ratio, (**c**) averaged Hamming score, and (**d**) averaged Hamming loss. The dots and the error bars indicate the mean and the standard deviation of the results from 10 different random seeds, respectively. To distinguish between UTH-BERT and NICT-BERT symbols, the UTH-BERT iteration number is offset by +0.1, and the NICT-BERT iteration number is offset by −0.1. The zeroth iteration shows the results from the pretrained SBERT systems.

## 3.2. Comparison between Search Engines

In this section, we compare the search accuracies of the search engine using the optimized SBERT with those of the search engines using the pretrained SBERT and Okapi BM25 as shown in Table 3; we found that the pretrained SBERT and Okapi BM25 were equally accurate. Using the trained SBERT system optimized by the iterative method described in the previous section, both the averaged exact match ratio and averaged Hamming score improved by more than 15 points. In addition, UTH-BERT performed two to three points better than NICT-BERT. Tables 4 and 5 show specific examples of search results for a specific query by search engines using the pretrained SBERT and the trained SBERT with UTH-BERT, respectively. Comparing the results in Tables 4 and 5, it is clear that the trained SBERT is more accurate.

**Table 3.** Comparison between search engines; bold type indicates the best results. The value of each evaluation metric is the mean and the standard deviation (SD) of the results from 10 different random seeds. The UTH-BERT and NICT-BERT results are derived from the fifth and sixth iterations in the iterative optimization, respectively.

| System | Models | Averaged Exact Match Ratio, Mean (SD) | Averaged Hamming Score, Mean (SD) | Averaged Hamming Loss, Mean (SD) |
|---|---|---|---|---|
| Okapi BM25 | UTH * | 0.264 (0.011) | 0.390 (0.012) | 0.0643 (0.0016) |
|  | NICT * | 0.268 (0.014) | 0.395 (0.012) | 0.0637 (0.0016) |
| Pretrained SBERT | UTH | 0.277 (0.016) | 0.412 (0.011) | 0.0612 (0.0012) |
|  | NICT | 0.251 (0.018) | 0.381 (0.011) | 0.0651 (0.0010) |
| Trained SBERT | UTH | **0.448 (0.018)** | **0.577 (0.016)** | **0.0421 (0.0019)** |
|  | NICT | 0.423 (0.020) | 0.548 (0.018) | 0.0458 (0.0022) |

* We did not use the BERT models, only their tokenizer for the Okapi BPM25 system.

**Table 4.** An example results of a similar text search by the PRETRAINED SBERT with the UTH-BERT model. The labels in red are those that match the labels in the query.

| | Summary Text of Medical Accident<br>Upper Row: Japanese Original Text; Lower Row: English Translation | Euclidean Distance | Category | Process |
|---|---|---|---|---|
| Query | 大動脈弁狭窄症の新生患児が、精査加療目的に、病院を紹介受診した。経皮的大動脈弁形成術（バルーン拡大術）施行中、体外式ペーシングカテーテルが破損し、破片が肺動脈末梢に残留した。回収処置試みるも、回収できず退院となった。患者家族から、病院としての今回の経過と今後の治療と検査についての文書を求めるというクレームがあった。一連の治療の適否が問題となった。有責。示談完了。賠償金の支払いあり。<br><br>A newborn child with aortic stenosis was referred to the hospital for close examination and treatment. During percutaneous aortic valvuloplasty (balloon enlargement), an external pacing catheter was broken, and a fragment remained in the peripheral pulmonary artery. Attempts were made to retrieve the catheter, but the debris could not be recovered, and the patient was discharged from the hospital. The patient's family complained to the hospital requesting documentation of this incident and future treatment and testing. The issue was the appropriateness of the treatments, the hospital was found liable, settlement was completed, and compensation was paid. | 0 | Internal Medicine | Surgery |
| First hit | 70歳代女性、狭心症で5 年前にPCI (冠動脈ステント留置術) 施行した患者が、冠動脈精査目的で、病院を受診した。心臓カテーテル検査施行中、右冠動脈解離が発生し、上行大動脈解離まで波及した。造影CT 施行し、緊急上行大動脈置換術ならびに右冠動脈バイパス術施行となった。患者代理人から、カテーテル手技の過失により、急性大動脈解離を発症し部分ろ部置換術及び冠動脈バイパス術を受け、20分程の検査のはずが、事故により10時間半に及ぶ手術となり精神的損害、術後の後遺障害を被ったというクレームがあった。一連の治療の適否が問題となった。動きなく長期経過し、終結申出あった。無責。賠償金の支払いなし。<br><br>A woman in her 70s who had undergone PCI (coronary artery stenting) five years ago for angina pectoris came to a hospital for a thorough examination of her coronary arteries. During cardiac catheterization, first the right coronary artery was dissected, and this then extended to the ascending aorta. Following contrast-enhanced CT, emergency ascending aortic replacement and right coronary artery bypass surgery were performed. The patient's attorney claimed that due to negligent catheterization, the patient developed an acute aortic dissection and underwent partial aortic replacement and coronary artery bypass surgery, which should have taken ~20 min. The issue was the suitability of her medical treatments. After a long period of time with no movement, the patient received a termination, but the hospital was found not liable, and no compensation was paid. | 1.8459 | Internal Medicine | Examination |
| Second hit | 40歳代男性、左陰嚢痛が主訴の患者が、精査加療を目的に、病院を受診した。造影CT 施行し、左尿管結石を認めた。経尿道的尿管結石除去術で硬膜外麻酔を施行も、うまく挿入できず、チューブを除去するもチューブの先端18 mm が体内に残存した。疼痛などの訴えあり。患者から、補償を希望するというクレームがあった。硬膜外チューブの体内遺残が問題となった。異物残置には過失があるとして有責と判断され、裁判外紛争解決 (ADR) で示談。有責。示談完了。賠償金の支払いあり。<br><br>A male patient in his 40s with a chief complaint of left scrotal pain visited a hospital for close examination and treatment. Contrast-enhanced CT confirmed a left ureteral stone. He received epidural anesthesia for transurethral ureteral stone removal, but the tube could not be inserted properly, and the 18 mm tip of the tube remained inside the body. The patient complained of pain and requested compensation. Leaving the foreign object behind in his body was considered negligent, and the hospital was found liable. This case was settled through alternative dispute resolution (ADR). Compensation was paid. | 1.8505 | Urology | Anesthetic |

**Table 4.** *Cont.*

| | Summary Text of Medical Accident<br>Upper Row: Japanese Original Text; Lower Row: English Translation | Euclidean Distance | Category | Process |
|---|---|---|---|---|
| Third hit | 80歳代女性、呼吸困難の患者が、精査加療目的に、病院を受診した。急性心不全の診断で、入院加療となった。ペースメーカー植え込み時、心房リードによる右心耳穿孔を原因とする心タンポナーデが発生した。心臓血管外科で手術の際、経皮的心肺補助装置 (PCPS) の脱血用カテーテルが腹部の下大静脈を穿刺し腹腔内へ出血したことが原因で死亡に至った。遺族から、技術的な失敗がなければ、元気に帰っていたかもしれないというクレームがあった。一連の治療の適否が問題となった。専門医は、迷入した時点でＣＴをとって手術していれば、死亡はしなかったとして、一部有責と結論付けた。有責。賠償金の支払いあり。<br><br>A female patient in her 80s with dyspnea visited a hospital for close examination and treatment. She was hospitalized for acute heart failure. During pacemaker implantation, cardiac tamponade occurred due to perforation of the right atrial appendage by atrial lead. During a cardiovascular surgery, a percutaneous cardiopulmonary support system (PCPS) debridement catheter perforated the inferior vena cava in the abdomen, causing bleeding into the abdominal cavity, which resulted in death. The bereaved family members complained that had it not been for the technical failure, the patient might have returned home in good health. The appropriateness of a series of treatments became an issue. The medical specialist concluded that the death would not have occurred if a CT had been taken and surgery had been performed at the time of the incorrect insertion and that the hospital was partially liable. Compensation was paid. | 1.8762 | <span style="color:red">Internal Medicine</span> | <span style="color:red">Surgery</span> |
| Fourth hit | 70歳代男性、顔面痙攣あり、近医で安定剤内服治療中の患者が、精査加療を目的に、病院を受診した。MRI 検査施行し、神経血管圧迫症候群と診断された。神経血管減圧術を行うことになった。術後合併症として硬膜下血腫、小脳出血、脳幹浮腫を生じ再手術するも意識状態改善なく死亡に至った。「医療事故調査委員会」の報告書に対する意見書とともに、「医療事故に対する補償金」の請求書が提出された。一連の治療の適否が問題となった。医師会は、合併症に対する診断処置の遅れがあり、有責と判断すると結論づけた。有責。賠償金の支払いあり。<br><br>A man in his 70s with facial spasm who was being treated with stabilizers by his primary care physician visited a hospital for close examination and treatment. An MRI scan was performed, and a diagnosis of neurovascular decompression syndrome was made. Neurovascular decompression surgery was performed. Subdural hematoma, cerebellar hemorrhage, and brainstem edema occurred as postoperative complications, and the patient died without improvement in consciousness despite reoperation. Along with the statement of opinion from the report of the Medical Adverse Event Investigation Committee, a claim for "compensation for medical accident" was submitted. The issue was the appropriateness of the treatments. The medical board found that there was a delay in diagnostic treatment for the complication and concluded that this hospital was liable. Compensation was paid. | 1.8835 | Neurosurgery | <span style="color:red">Surgery</span> |
| Fifth hit | 50歳代女性、胸部の不快感が主訴の患者が、精査加療を目的に、病院の救急外来を受診した。大動脈弁閉鎖不全症の診断で大動脈弁置換術施行したが、術中、機械弁のミスマッチで再置換を行い、左前下行枝領域に広範な心筋梗塞を合併し、大動脈バルーンパイピング使用、冠動脈バイパス術を行った。術後2 日目に心機能維持できず、PCPS を装着し、補助人工心臓装着のため、大学病院へ転院となったが、全身状態、心機能の回復が得られず、急性心筋梗塞により死亡した。事故調査委員会が開催され、予定外に大動脈遮断時間が長くなったこと、剖検例で発覚した左前下行枝領域の狭窄病変が存在したこと、遮断解除後の循環動態の管理などが左前下行枝領域の心機能低下から心筋梗塞へと進行した要因となった可能性が考えられ、診療体制の問題点としては、心臓血管外科チームがうまく構成されていなかったことがあげられるという結論となった。遺族から、大学病院搬送時の遅延について問題があったというクレームがあった。一連の治療の適否が問題となった。専門医は、死亡は大動脈弁置換術に伴う周術期心筋梗塞と思われ、大動脈弁置換術と死亡とは因果関係があると結論付けた。無責。請求取り下げ。賠償金の支払いなし。 | 1.8997 | Surgery | <span style="color:red">Surgery</span> |

**Table 4.** *Cont.*

| | Summary Text of Medical Accident<br>Upper Row: Japanese Original Text; Lower Row: English Translation | Euclidean Distance | Category | Process |
|---|---|---|---|---|
| Fifth hit | A female patient in her 50s with a chief complaint of chest discomfort visited a hospital emergency department for close examination and treatment. She underwent aortic valve replacement with a diagnosis of aortic regurgitation. Intraoperatively, she underwent revision due to a mismatch of the mechanical valve, which was complicated by extensive myocardial infarction in the left anterior descending branch region, and she underwent aortic balloon piping and coronary artery bypass surgery. On the second postoperative day, cardiac function could not be maintained, and the patient was placed on PCPS and transferred to a university hospital for placement of an artificial heart. However, her general condition and cardiac function did not recover, and she died of acute myocardial infarction. An accident investigation committee was convened that concluded that the unscheduled prolonged aortic blockade time, the presence of a stenotic lesion in the left anterior descending branch region that was discovered in the autopsy case, and the management of circulatory dynamics after unblocking may have contributed to the progression of the left anterior descending branch region from a loss of cardiac function to myocardial infarction. The conclusion was that the cardiovascular surgery team was not well organized. The family complained that there was a problem with the delay in transporting the patient to the university hospital. The issue was the appropriateness of the treatments. The medical specialist concluded that the death appeared to be a perioperative myocardial infarction associated with aortic valve replacement and that there was a causal relationship between aortic valve replacement and death. This hospital was not liable. The claim was withdrawn. No compensation was paid. | 1.8997 | Surgery | Surgery |
| Sixth hit | 70歳代男性、前立腺肥大症の患者が、精査加療目的に、病院を受診した。術前の検査から、重度の僧帽弁閉鎖不全症で手術適応と診断された。低侵襲僧帽弁形成術(MICS-MVP) 施行後、血圧高値で薬剤投与するもコントロールできず、術後低心拍出症候群(LOS) となり、再開胸止血術施行した。4時間後、血行動態不安定で、正中小切開で心タンポナーデを認めた。3回目の手術で再開胸( 正中切開)止血術で縫合止血し、重篤な合併症なく経過後、退院となった。患者から、血圧が上昇したのは、医療過誤ではないかなどのクレームがあった。一連の治療の適否が問題となった。後遺障害診断書を作成した。有責。示談完了。和解金の支払、いあり。<br><br>A man in his 70s with benign prostatic hyperplasia visited a hospital for close examination and treatment. Preoperative examination revealed that he had severe mitral regurgitation and was indicated for surgery. After minimally invasive mitral valvuloplasty (MICS-MVP), the patient's blood pressure was elevated and uncontrolled despite drug administration, resulting in postoperative low cardiac output syndrome (LOS). 4 h later, the patient was hemodynamically unstable and had cardiac tamponade through a median incision. After the third operation, he was discharged from the hospital after suture hemostasis was achieved through a reopened chest (median incision) hemostasis without serious complications. The patient complained that the elevated blood pressure was medical malpractice. The appropriateness of a series of treatments became an issue. A permanent disability certificate was prepared. This case was liable. Settlement was completed. Compensation was paid. | 1.9071 | Surgery | Surgery |
| Seventh hit | 50歳代女性、左足趾疼痛とチアノーゼ出現で前医受診した患者が、精査加療を目的に、病院を受診した。精査の結果、骨盤内腫瘤による左腸骨動脈狭窄と腸骨静脈圧迫によるうっ血と診断された。両側付属器切除術希望で手術施行も、連絡ミスにより左卵巣残存となった。患者から、卵巣残存に起因する病気リスクへの不安、それに伴う検診費用などについて賠償請求するというクレームがあった。一連の治療の適否が問題となった。医賠研は、院内の連絡ミスによる臓器の取り残しでミスは明らかと結論づけた。有責。示談完了。賠償金の支払いあり。<br><br>A woman in her 50s, who had been seen by her previous physician for left toe pain and cyanosis, visited a hospital for close examination and treatment. After close examination, she was diagnosed as having left iliac artery stenosis due to a pelvic mass and congestion due to compression of the iliac vein. The patient requested bilateral adnexectomy and underwent surgery, but due to a miscommunication, the left ovary remained. The patient claimed that she was concerned about the risk of disease due to the residual ovary and claimed compensation for the associated cost of medical examinations. The appropriateness of a series of treatments became an issue. A consultant physician concluded that the mistake was obvious, as the organs were left behind due to miscommunication within the hospital. This case was liable. Settlement was completed. Compensation was paid. | 1.9101 | Obstetrics and Gynecology | Surgery |

**Table 4.** *Cont.*

| | **Summary Text of Medical Accident**<br>**Upper Row: Japanese Original Text; Lower Row: English Translation** | **Euclidean Distance** | **Category** | **Process** |
|---|---|---|---|---|
| Eighth hit | 60歳代女性、持続する胸痛が主訴の患者が、精査加療目的に、病院の救急外来を受診した。不安定狭心症の診断で緊急冠動脈造影検査及び経皮的冠動脈ステント留置術施行した。手術に伴い血腫が発生、皮膚びらん、水疱形成が見られた。患者から、病気を作られたなど医療ミスを主張するクレームがあった。一連の治療の適否が問題となった。病院は合併症の姿勢。カルテ開示請求あったが、動きなく事案終了。再燃した場合は事案復活。無責。賠償金の支払いなし。<br><br>A female patient in her 60s with a chief complaint of persistent chest pain visited the emergency department of a hospital for close examination and treatment. She underwent emergency coronary angiography and percutaneous coronary stenting with a diagnosis of unstable angina pectoris. Haematoma, skin erosion, and blister formation were observed following the surgery. The patient made a complaint alleging medical malpractice, such as having been made ill. The appropriateness of a series of treatments became an issue. The hospital took a complication stance. There was a request for disclosure of medical records, but the case was closed without action. In the case of relapse, this case was reinstated. The case is not liable. No compensation was paid. | 1.9495 | Internal Medicine | Surgery |
| Ninth hit | Wilson病の女児患者が、肝移植を目的に、入院した。中心静脈ライン挿入時に右鎖骨下動脈を穿刺した。また、生体肝移植手術後に抜管した際に右鎖骨下動脈から多量出血した。患者家族より通知書届き、上記の施術により血気胸、脳実質障害、高次脳機能障害を負ったとして損害賠償請求があった。高次脳機能障害と大量出血の因果関係が問題となった。調停となった。有責。示談完了。賠償金の支払いあり。<br><br>A girl patient with Wilson's disease was admitted to a hospital for liver transplantation. The right subclavian artery was punctured during the insertion of a central venous line. She also suffered massive bleeding from the right subclavian artery during extubation after living donor liver transplantation. The patient's family claimed damages for hemopneumothorax, brain parenchymal disorder, and higher brain dysfunction caused by the above procedure. The causal relationship between the higher brain dysfunction and the massive hemorrhage became an issue. This case went to arbitration. The case was liable. Settlement was completed. Compensation was paid. | 1.9499 | Unknown | Surgery |
| Tenth hit | 60歳代男性、単独交通事故受傷後、他施設入院していた患者、左腎嚢胞出血及び左腎結石の精査加療を目的に、病院を受診した。経尿道的尿管結石破砕術施行した。術中尿管損傷し、開腹して尿管端々吻合術にて修復した。患者から、予期せぬ開腹術に対して慰謝料請求するというクレームがあった。一連の治療の適否が問題となった。専門医は、患者の症状から手術適応に疑問があり、対外衝撃波結石破砕術(ESW) の選択をする必要があったのではないかと結論付けた。有責。示談完了。賠償金の支払いあり。<br><br>A man in his 60s, who had been hospitalized at another facility after a single traffic accident injury, visited the hospital for close examination and treatment of a left renal cyst hemorrhage and left renal calculus. He underwent transurethral ureteral lithotripsy. The ureter was injured intraoperatively and was repaired by laparotomy with ureteral end anastomosis. The patient claimed for compensation for the unexpected laparotomy. The appropriateness of a series of treatments became an issue. The specialist concluded that the patient's symptoms called into question the indication for surgery and that external shock wave lithotripsy (ESW) should have been the treatment of choice. This case is liable. Settlement was completed. Compensation was paid. | 1.9531 | Urology | Surgery |

Exact match ratio: 0.200; Hamming score: 0.433; Hamming loss: 0.0529.

**Table 5.** An example results of a similar text search by the TRAINED SBERT with the UTH BERT model. The query is the same as in Table 4. The labels in red are those that match the labels in the query.

| | Summary Text of Medical Accident<br>Upper Row: Japanese Original Text; Lower Row: English Translation | Euclidean Distance | Category | Process |
|---|---|---|---|---|
| Query | 大動脈弁狭窄症の新生患児が、精査加療目的に、病院を紹介受診した。経皮的大動脈弁形成術 ( バルーン拡大術) 施行中、体外式ペーシングカテーテルが破損し、破片が肺動脈末梢に残留した。回収処置試みるも、回収できず退院となった。患者家族から、病院としての今回の経過と今後の治療と検査についての文書を求めるというクレームがあった。一連の治療の適否が問題となった。有責。示談完了。賠償金の支払いあり。<br><br>A newborn child with aortic stenosis was referred to the hospital for close examination and treatment. During percutaneous aortic valvuloplasty (balloon enlargement), an external pacing catheter was broken, and a fragment remained in the peripheral pulmonary artery. Attempts were made to retrieve the catheter, but the debris could not be recovered, and the patient was discharged from the hospital. The patient's family complained to the hospital requesting documentation of this incident and future treatment and testing. The appropriateness of a series of treatments became an issue. This case was liable. Settlement was completed. Compensation was paid. | 0 | <span style="color:red">Internal Medicine</span> | <span style="color:red">Surgery</span> |
| First hit | 70歳代女性、狭心症で5年前にPCI( 冠動脈ステント留置術) 施行した患者が、冠動脈精査目的で、病院を受診した。心臓カテーテル検査施行中、右冠動脈解離が発生し、上行大動脈解離まで波及した。造影CＴ 施行し、緊急上行大動脈置換術ならびに右冠動脈バイパス術施行となった。患者代理人から、カテーテル手技の過失により、急性大動脈解離を発症し部分ろ部置換術及び冠動脈バイパス術を受け、20分程の検査のはずが、事故により10時間半に及ぶ手術となり精神的損害、術後の後遺障害を被ったというクレームがあった。一連の治療の適否が問題となった。動きなく長期経過し、終結申出あった。無責。賠償金の支払いなし。<br><br>A woman in her 70s, who had undergone PCI (coronary artery stenting) 5 years ago for angina pectoris, came to a hospital for a thorough examination of her coronary arteries. During cardiac catheterization, dissection of the right coronary artery occurred, which spread to dissection of the ascending aorta. Contrast-enhanced CT was performed, and emergency ascending aortic replacement and right coronary artery bypass surgery were performed. The patient's attorney claimed that due to negligent catheterization procedures, the patient developed an acute aortic dissection and underwent partial aortic replacement and coronary artery bypass surgery, which should have taken ~20 min. The suitability of a series of medical treatments became an issue. After a long period of time with no movement, a termination offer was made. This case is not liable. No compensation was paid. | 3.5913 | <span style="color:red">Internal Medicine</span> | Examination |
| Second hit | 70歳代男性、急性心筋梗塞、胆石胆嚢炎の患者が、手術を目的に入院した。CTで総胆管結石認め、内視鏡的総胆管切石術を施行されたが、胆管損傷により胆汁性腹膜炎を発症した。直ちに胆のう摘出術及び腹腔ドレナージを施行され、一時改善も直腸潰瘍からの下血により出血性ショックとなり、死亡した。内視鏡的総胆管切石術の手技が問題となった。専門医は手技に医療上の問題点は認められず、合併症であると判断した。損傷の程度から合併症と言い切れるか等、微妙であると判断した。有責。示談完了。賠償金の支払いあり。<br><br>A male patient in his 70s with acute myocardial infarction and cholelithiasis was admitted for surgery. The patient underwent immediate cholecystectomy and abdominal drainage, and although he temporarily improved, he went into hemorrhagic shock due to bleeding from a rectal ulcer and died. The technique of endoscopic choledocholithotomy was problematic. The specialist found no medical problems with the procedure and concluded that it was a complication. He determined that the extent of the injury was too subtle to be considered a complication, etc. This case is liable. Settlement was completed. Compensation was paid. | 4.0901 | <span style="color:red">Internal Medicine</span> | Treatment |
| Third hit | 80歳代女性、呼吸困難の患者が、精査加療目的に、病院を受診した。急性心不全の診断で、入院加療となった。ペースメーカー植え込み時、心房リードによる右心耳穿孔を原因とする心タンポナーデが発生した。心臓血管外科で手術の際、経皮的心肺補助装置(PCPS) の脱血用カテーテルが腹部の下大静脈を穿孔し腹腔内へ出血したことが原因で死亡に至った。遺族から、技術的な失敗がなければ、元気に帰っていたかもしれないというクレームがあった。一連の治療の適否が問題となった。専門医は、迷入した時点でCＴ をとって手術していれば、死亡はしなかったとして、一部有責と結論付けた。有責。賠償金の支払いあり。 | 4.4096 | <span style="color:red">Internal Medicine</span> | <span style="color:red">Surgery</span> |

**Table 5.** *Cont.*

| | Summary Text of Medical Accident<br>Upper Row: Japanese Original Text; Lower Row: English Translation | Euclidean Distance | Category | Process |
|---|---|---|---|---|
| Third hit | A female patient in her 80s with dyspnea visited a hospital for close examination and treatment. She was hospitalized for acute heart failure. During pacemaker implantation, cardiac tamponade occurred due to perforation of the right atrial appendage by atrial lead. During a cardiovascular surgery, a percutaneous cardiopulmonary support system (PCPS) debridement catheter perforated the inferior vena cava in the abdomen, causing bleeding into the abdominal cavity, which resulted in death. The bereaved family members complained that had it not been for the technical failure, the patient might have returned home in good health. The appropriateness of a series of treatments became an issue. The medical specialist concluded that the death would not have occurred if a CT had been taken and surgery had been performed at the time of the incorrect insertion and that it was partially liable. Compensation was paid. | 4.4096 | Internal Medicine | Surgery |
| Fourth hit | 70歳代男性、心室頻拍の患者が精査加療目的に病院を受診した。カテーテルアブレーション後、心タンポナーデを機に心原性ショックとなり、冠攣縮性狭心症による心筋虚血状態となった。心停止から蘇生後脳症となった。患者家族から、手術を受けなければこのような合併症が起こらなかったのではないかというクレームがあった。一連の治療の適否が問題となった。提訴され、裁判では、主に医療の適否が問題となった。有責。第一審和解。和解金の支払いあり。<br><br>A male patient in his 70s with ventricular tachycardia visited a hospital for close examination and treatment. After catheter ablation, he went into cardiogenic shock due to cardiac tamponade and myocardial ischemia due to coronary angina pectoris. After resuscitation from cardiac arrest, the patient became encephalitic. The patient's family complained that such complications would not have occurred had the patient not undergone surgery. The appropriateness of a series of treatments became an issue. A lawsuit was filed, and at trial, the main issue was the adequacy of medical care. This case was liable and settled in the first instance. Settlement was paid. | 4.4647 | Internal Medicine | Surgery |
| Fifth hit | 40歳代男性、動悸が主訴の患者が、精査加療を目的に、病院を紹介受診した。心室頻拍症のためカテーテルアブレーション(ABL) 施行し、一定の効果が得られたことを確認し退院した。上室性頻拍が再出現し、房室結節回帰性頻拍に対し、再度ABL 施行中、完全房室ブロックを合併した。房室ブロックの改善見られず、恒久的ペースメーカーの埋込手術施行となった。患者から、労働の制限が掛かって失職した場合は補償してもらえるのかなどのクレームがあった。カルテ開示請求後、訴訟となった。2度目ABL の房室ブロックの認識時点について問題となった。専門医は2秒と3秒の時点で房室ブロックを認識し通電を中止すれば恒久的な房室ブロックにはならなかった可能性が高いとして問題があったと結論付けた。提訴され、裁判では、アブレーション中の異常波形出現時点が問題となった。裁判所は、和解勧告した。有責。第一審和解。賠償金の支払いあり。<br><br>A male patient in his 40s with a chief complaint of palpitations was referred to a hospital for close examination and treatment. He underwent catheter ablation (ABL) for ventricular tachycardia and was discharged after confirming that the procedure was effective. Supraventricular tachycardia reappeared, and during another ABL for atrioventricular nodal regression tachycardia, complete atrioventricular block occurred. The atrioventricular block did not improve, and the patient underwent surgery for implantation of a permanent pacemaker. The patient complained that he was restricted from working and wondered if he would be compensated if he lost his job. After a request for disclosure of medical records, a lawsuit was filed. The specialist concluded that there was a problem because if he had recognized the AV block at the 2- and 3-s time points and stopped the energization, there was a good chance that permanent AV block would not have occurred. The case was filed, and the trial court took issue with the point of appearance of the abnormal waveform during the ablation. The court recommended settlement. This case was liable and settled in the first instance. Compensation was paid. | 4.6217 | Internal Medicine | Surgery |
| Sixth hit | 80歳代男性、労作時息切れが主訴の患者が、精査加療目的に、病院を受診した。完全房室ブロックによるうっ血性心不全の診断で入院となった。左鎖骨下ペースメーカー植え込み術(PMI) 施行した。PMI 施行時、創部からの止血困難でドレーン挿入となった。創部壊死・デバイス感染防止目的にデバイス・リード抜去術を施行した。右側へ再PMI 施行した。患者家族から、不必要な手術費用、延伸した入院費等の補償を求めるというクレームがあった。一連の治療の適否が問題となった。ポケット作成時の筋肉組織の破断回避操作及び、閉創時の出血部位・量の観察が不十分であったことから、創部が壊死し、再手術、入院期間が延長することになったとして、見舞金が支払われた。無責。示談完了。見舞金の支払いあり。 | 4.7687 | Internal Medicine | Surgery |

**Table 5.** *Cont.*

| | Summary Text of Medical Accident<br>Upper Row: Japanese Original Text; Lower Row: English Translation | Euclidean Distance | Category | Process |
|---|---|---|---|---|
| Sixth hit | A male patient in his 80s with a chief complaint of shortness of breath on exertion came to a hospital for close examination and treatment. He was admitted to the hospital with a diagnosis of congestive heart failure due to complete atrioventricular block. A left subclavian pacemaker implantation (PMI) was performed, and a drain was inserted due to difficulty in stopping bleeding from the wound during PMI. Device and lead removal was performed to prevent wound necrosis and device infection. Re-PMI was performed on the right side. The patient's family complained about unnecessary surgical costs and extended hospitalization, demanding compensation. The appropriateness of a series of treatments became an issue. The patient was awarded compensation for the avoidance of rupture of muscle tissue during pocket creation and inadequate observation of the site and amount of bleeding at the time of wound closure, which resulted in wound necrosis, reoperation, and prolonged hospitalization. This case was not liable. Settlement was completed. Sympathetic compensation was paid. | 4.7687 | Internal Medicine | Surgery |
| Seventh hit | 60歳代女性、狭心症の患者が、精査加療を目的に、病院を受診した。冠動脈CT施行され、狭窄が見つかりカテーテル検査施行した。心筋前壁広範囲梗塞を合併し心破裂で死亡した。事故調査委員会が開催され、カテーテル適応あり、心破裂の予知は難しく、合併症の範囲であるという結論となった。遺族から、簡単なステント手術で死亡したのは納得できないというクレームがあった。一連の治療の適否が問題となった。第三者委員会の結論から無責通知、通知後動き無く終了となった。無責。示談完了。賠償金の支払いなし。<br><br>A woman in her 60s with angina pectoris visited a hospital for close examination and treatment. She underwent a coronary CT scan, which revealed stenosis, and catheterization was performed. The patient died of cardiac rupture complicated by extensive infarction of the anterior myocardial wall. An accident investigation committee was held and concluded that catheterization was indicated, that it was difficult to predict cardiac rupture, and that it was within the scope of complications. The family complained that they were not satisfied that a simple stent procedure had caused death. The appropriateness of a series of treatments became an issue. The third-party committee concluded that there was no liability, and the case was closed without further action. The case was not liable. Settlement was completed. No compensation was paid. | 4.9096 | Internal Medicine | Surgery |
| Eighth hit | 70歳代男性、無痛性心筋虚血の患者が、精査を目的に、病院を受診した。経皮的冠動脈形成術(PCI) 施行後、問題無く経過していたが、急に血圧低下あり、心停止し、心肺蘇生開始した。心タンボナーデ認め、緊急心カテPCI 施行した。1回目のPIC 施行部に穿孔認め、ステント追加留置した。蘇生後、低酸素脳症発症し遷延性意識障害となり、敗血症により死亡した。遺族から、診療記録の写し、事故調査の実施と説明を求めるというクレームがあった。経過の適否が問題となった。医賠研は、PCI 施行部の穿孔自体はローターブレーターによる合併症で過失とは言えないが、脳への血行不十分とし低酸素脳症を起こしたことは過失と評価されるだろうと結論付けた。提訴され、裁判では、主に手術適応や手技が問題となった。有責。第一番和解。賠償金の支払いあり。<br><br>A 70-year-old male with painless myocardial ischemia came to a hospital for a thorough examination. After percutaneous coronary angioplasty (PCI), the patient was doing well, but suddenly his blood pressure dropped, and he had a cardiac arrest. The patient had a perforation at the site of the first PCI, and an additional stent was implanted. After resuscitation, hypoxic encephalopathy developed, resulting in prolonged loss of consciousness, and the patient died of sepsis. The bereaved family complained, demanding a copy of the medical records, an investigation of the accident, and an explanation of the incident. The appropriateness of the process was in question. A consultant physician concluded that the perforation of the PCI site itself was a complication from the rotablator and could not be considered negligence, but that the lack of blood circulation to the brain and the hypoxic encephalopathy that resulted could be evaluated as negligence. This case was filed, and the main issues at trial were surgical indications and technique. This case was liable and settled in the first instance. Compensation was paid. | 4.9403 | Internal Medicine | Surgery |

**Table 5.** *Cont.*

| | Summary Text of Medical Accident<br>Upper Row: Japanese Original Text; Lower Row: English Translation | Euclidean Distance | Category | Process |
|---|---|---|---|---|
| Ninth hit | Wilson病の女児患者が、肝移植を目的に、入院した。中心静脈ライン挿入時に右鎖骨下動脈を穿刺した。また、生体肝移植手術後に抜管した際に右鎖骨下動脈から多量出血した。患者家族より通知書届き、上記の施術により血気胸、脳実質障害、高次脳機能障害を負ったとして損害賠償請求があった。高次脳機能障害と大量出血の因果関係が問題となった。調停となった。有責。示談完了。賠償金の支払いあり。<br><br>A girl patient with Wilson's disease was admitted to a hospital for liver transplantation. The right subclavian artery was punctured during the insertion of a central venous line. She also suffered massive bleeding from the right subclavian artery during extubation after living donor liver transplantation. The patient's family claimed damages for hemopneumothorax, brain parenchymal disorder, and higher brain dysfunction caused by the above procedure. The causal relationship between the higher brain dysfunction and the massive hemorrhage became an issue. This case went to arbitration. The case was liable. Settlement was completed. Compensation was paid. | 5.1120 | Unknown | Surgery |
| Tenth hit | 70歳代男性、胆石症疑いの患者が、精査加療を目的に、病院を紹介受診した。内視鏡的胆管結石排石術施行し、術中に出血した。翌日血管塞栓術施行し、退院後他院紹介となった。患者弁護士から治療に対する不法行為又は責務不履行責任として証拠保全申立あった。治療経過に過失があったか問題となった。専門医は、手技ミスはなく合併症であると判断し、問題がなかったとした。無責回答後動き無く立ち消え。無責。示談完了。賠償金の支払いなし。<br><br>A male patient in his 70s with suspected cholelithiasis was referred to a hospital for close examination and treatment. Endoscopic choledocholithotomy was performed, and bleeding occurred during the procedure. The next day, he underwent a vascular embolization procedure and was referred to another hospital after discharge. The patient's attorney filed a motion for the preservation of evidence for tortious behavior or breach of duty in relation to the treatment. The question arose as to whether there was negligence during treatment. The specialist determined that there was no procedural error, but rather a complication, and that there was no problem. After the no-fault answer, the case went away without any motion. This case was not liable. Settlement was completed. No compensation was paid. | 5.1246 | Internal Medicine | Surgery |

Exact match ratio: 0.700; Hamming score: 0.800; Hamming loss: 0.0176.

## 4. Discussion

The purpose of this study was to develop an information retrieval system to find similar accidents to a specific medical accident with high accuracy and high speed for a Japanese database of 1165 closed medical malpractice claims. In the database, we manually created a short text summarizing each accident and assigned each a category and a process (e.g., orthopedics and surgery or plastic surgery and follow-up). We developed the semantic search engines based on SBERT, which was reported to have excellent speed and accuracy in extracting similar texts [14]. The search engines extracted 10 similar summary texts that were close to a given query in the Euclidean distance of the embedding. To train SBERT, we conducted iterative optimization to efficiently expand the training dataset and gradually increase the retrieval accuracy. We defined three evaluation metrics that indicated how well the labels of the 10 extracted similar instances matched the labels of the query: averaged exact match ratio, averaged Hamming score, and averaged Hamming loss. For the SBERT encoder, we employed UTH-BERT, the medical BERT model pretrained on clinical records, and NICT-BERT, a generic BERT model pretrained on Japanese Wikipedia articles, and we compared the results from these models. We also built search engines based on lexical similarity using Okapi BM25 for comparison.

In the early stages of the iterative optimization, the number of label-matched pairs increased with the number of iterations, and the search accuracy improved as expected. This could mean that the SBERT model gradually adapted to the style of texts in our database through the iterative optimization while updating the dataset. The evaluation metrics were highest around the middle five of the ten iterations, after which there was a gradual decline. The reason the search accuracy did not monotonically increase with

the number of iterations could have been duplicated label-matched pairs in the dataset and/or over-training on the training dataset. The current implementation allowed for duplicating the pairs created when the query was 'A' and its similar text was 'B' $(s_A, s_B)$ and pairs created when the query was 'B' and its similar text was 'A' $(s_B, s_A)$. Duplications increased with each optimization iteration, which might have prevented accuracy from improving. In the future, the iterative optimization algorithm should be modified to prevent this duplication.

The search engine using the best SBERT obtained via iterative optimization showed higher search accuracy than the search engine using Okapi BM25 using both UTH-BERT and NICT-BERT as encoders. In both the study on detecting COVID-19 misinformation on social media [23] and the study on unsupervised FAQ retrieval with question generation [24], the information retrieval systems using BERT are 15 to 20 points more accurate for mean reciprocal rank (MRR) than those using BM25. Our results are consistent with these previous studies in that BERT performs better than BM25.

In the search engine using trained SBERT, the system using UTH-BERT as the encoder showed a higher search accuracy than the one using NICT-BERT as the encoder. In a study on extracting pain expression from medical records using UTH-BERT [25], UTH-BERT showed ~10 points higher accuracy in the F1-score evaluation than Kyoto University BERT, which was pretrained on Japanese Wikipedia articles. Our result is consistent with the previous study's finding in that UTH-BERT performed better than the general BERT models for medical texts. We consider that the higher retrieval accuracy of UTH-BERT than of NICT-BERT is largely owing to the accuracy of morphological analysis. To give an example, 犬吠様咳嗽 (the Japanese expression *kennel cough*) became a single token with the UTH-BERT tokenizer, where the vocabulary was created from the corpus of clinical records. In contrast, the NICT-BERT tokenizer split 犬吠様咳嗽 into four tokens, 犬 (*dog*), 吠様 (*barking-like*), 咳 (*cough*), and 嗽 (*gargling*), which have somewhat different meanings.

As described above, a similar document retrieval system using the trained SBERT demonstrated high retrieval accuracy in our closed-claims database. Currently, we are working on expanding our database, which we expect to eventually have more than 30,000 cases. By applying the search system we established in this study, we expect to be able to utilize the expanded closed-claims database more effectively when extracting similar medical accidents.

**Author Contributions:** Conceptualization, N.F., Y.O. and S.K.; methodology, N.F.; software, N.F.; validation, N.F., Y.O. and S.K.; formal analysis: N.F. and Y.O.; data curation, N.F. and Y.O.; writing—original draft preparation, N.F.; writing—review and editing, N.F., Y.O. and S.K.; visualization, N.F.; supervision, S.K.; project administration, Y.O.; funding acquisition, N.F. and Y.O. All authors have read and agreed to the published version of the manuscript.

**Funding:** This work was supported by JSPS KAKENHI Grant Numbers JP18KT0099, JP19K10542, and JP21K10359.

**Institutional Review Board Statement:** This study complied with the Japanese epidemiologic study guidelines and was approved by the ethics committee of our university: Teikyo University Ethical Review Board for Medical and Health Research Involving Human Subjects (authorized number Teirin19-059). To preserve anonymity, no short texts or labels created in this study contained information that could identify specific individuals or organizations. Therefore, the ethics committee waived the requirement for informed consent in this retrospective study.

**Informed Consent Statement:** Not applicable.

**Data Availability Statement:** All relevant data are within the paper. Researchers need to contact the author, Yasuhiro Otaki, for access to the original summary text and the labels of each medical accident.

**Acknowledgments:** We would like to thank Hanaoka, the assistant, for her help in writing the short text summaries and annotating. We would like to thank Tatsuji Katsuki for his help in data curation.

**Conflicts of Interest:** The authors declare no conflict of interest.

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
