# Peer review of "Accuracy of the Sentence-BERT Semantic Search System for a Japanese Database of Closed Medical Malpractice Claims"

_applsci, doi:10.3390/app13064051_

Round 1

Reviewer 1 Report

This is an extremely well written paper with one small exception which I note below.  For the most part, I am happy with the content.  Being an AI person, though, I was disappointed that the authors do not spend any time addressing the algorithms behind the BERT machine learning approach or the bi-encoder versus cross-encoder approaches.  I think that anyone who reads this might question how these work and why these selections were made.  If the authors have room in this paper, I would encourage them to add at least a couple of paragraphs if not a full subsection talking about the algorithms themselves.

The only English issue I found was this sentence in the abstract:  We assigned each case in the database a short Japanese summary of the accident as well as two labels: category classified the hospital department, medical profession, and property damages, and process indicated a failed medical procedure.

I couldn't parse this because it says "two labels" but the first label appears to be of three values (department, profession, damages), and the expression "category classified the hospital department" doesn't make sense either.  Is this "the hospital department responsible for classifying the type of claim" perhaps?

Two minor things that I encourage the authors to modify.  The first is that you said you wrote short texts for each of the 1165 claims and also annotated each by category and process.  How long did this step take?  And I thought table 2 had some terminology that needed more explanation such as the role of the random seed and the maximum sequence length (and what the +3 meant in that entry). 

I might also recommend either moving the content in the appendices into the paper proper, or at least presenting a partial case as provided in the appendix in section 3.2 or 4. 

For the most part, this is a good paper but I think readers might be disappointed that far more effort was taken to describe the evaluation metrics (section 2.4) than the AI behind the BERT-used machine learning algorithms.  Or maybe that's just me.

Reviewer 2 Report

I have the following suggestions:

---There are the latest versions of BERT such as S-BERT and other similar or better methods available within Deep neural net. So Why BERT?

--- Sample sizes selected are quite small or it do not assure reliability and consistency of the method.

---Can not find proper conclusion 

--- Related works must be improved and contribution and findings within related works must be discussed.
